# Limited Evidence for Microbial Transmission in the Phylosymbiosis between Hawaiian Spiders and Their Microbiota

Benoît Perez-Lamarque,[a,b] Henrik Krehenwinkel,[c] Rosemary G. Gillespie,[d] Hélène Morlon[a]

[a]Institut de Biologie de l'ENS, École Normale Supérieure, CNRS, INSERM, Université PSL, Paris, France
[b]Institut de Systématique, Évolution, Biodiversité, Muséum national d'Histoire naturelle, CNRS, Sorbonne Université, EPHE, Université des Antilles, Paris, France
[c]Department of Biogeography, Trier University, Trier, Germany
[d]Department of Environmental Science, Policy and Management, University of California, Berkeley, California, USA

**ABSTRACT** The degree of similarity between the microbiotas of host species often mirrors the phylogenetic proximity of the hosts. This pattern, referred to as phylosymbiosis, is widespread in animals and plants. While phylosymbiosis was initially interpreted as the signal of symbiotic transmission and coevolution between microbes and their hosts, it is now recognized that similar patterns can emerge even if the microbes are environmentally acquired. Distinguishing between these two scenarios, however, remains challenging. We recently developed HOME (host-microbiota evolution), a cophylogenetic model designed to detect vertically transmitted microbes and host switches from amplicon sequencing data. Here, we applied HOME to the microbiotas of Hawaiian spiders of the genus *Ariamnes*, which experienced a recent radiation on the archipelago. We demonstrate that although Hawaiian *Ariamnes* spiders display a significant phylosymbiosis, there is little evidence of microbial vertical transmission. Next, we performed simulations to validate the absence of transmitted microbes in *Ariamnes* spiders. We show that this is not due to a lack of detection power because of the low number of segregating sites or an effect of phylogenetically driven or geographically driven host switches. *Ariamnes* spiders and their associated microbes therefore provide an example of a pattern of phylosymbiosis likely emerging from processes other than vertical transmission.

**IMPORTANCE** How host-associated microbiotas assemble and evolve is one of the outstanding questions of microbial ecology. Studies aiming at answering this question have repeatedly found a pattern of "phylosymbiosis," that is, a phylogenetic signal in the composition of host-associated microbiotas. While phylosymbiosis was often interpreted as evidence for vertical transmission and host-microbiota coevolution, simulations have now shown that it can emerge from other processes, including host filtering of environmentally acquired microbes. However, distinguishing the processes driving phylosymbiosis in nature remains challenging. We recently developed a cophylogenetic method that can detect vertical transmission. Here, we applied this method to the microbiotas of recently diverged spiders from the Hawaiian archipelago, which display a clear phylosymbiosis pattern. We found that none of the bacterial operational taxonomic units is vertically transmitted. We show with simulations that this result is not due to methodological artifacts. Thus, we provide a striking empirical example of phylosymbiosis emerging from processes other than vertical transmission.

**KEYWORDS** microbiota, phylosymbiosis, vertical transmission, host filtering, Hawaiian arthropods, endosymbionts

**M**ost multicellular organisms host complex microbial communities, referred to as microbiotas, which provide important functions to their hosts (1, 2). Although these microbial communities can fluctuate over short timescales and according to external variables such as animal diet or soil composition (3–5), microbiotas of host individuals from the

Address correspondence to Benoît Perez-Lamarque, benoit.perez@ens.psl.eu.

The authors declare no conflict of interest.

same species often tend to be more similar than microbiotas from different host species (6). Over long timescales, the extent to which these microbiotas, and the functions they provide to their hosts, are conserved will depend on the relative tendency of microbes to colonize host individuals at each generation (7), which is influenced by their modes of inheritance. At the two extremes, microbes can be either transmitted from generation to generation (vertical transmission, e.g., directly through the maternal germ line or indirectly by social contacts with relatives from the same host lineage [8, 9]) or acquired from the environment during the lifetime of each host individual independently of the previous host generation (environmental acquisition) (9–11). In the latter case, the maintenance of the microbes from host generation to host generation depends on their abundance in the host environment and their availability to colonize the host niche, which can be seen as an ecological filter (12).

Irrespective of how a host acquires its microbiota, microbiotas of closely related host species are often more similar than those of distantly related species, such that host dendrograms constructed from the similarities of whole microbiota communities tend to mirror the host phylogeny (13, 14). This pattern, referred to as phylosymbiosis, has for example been documented for the gut microbiotas of primates (15, 16) and arthropods (17) and in the roots of plants (18). However, how and why this pattern emerges has been intensively debated (12, 19). In particular, phylosymbiosis is expected to emerge if some specific microbial lineages are vertically transmitted during host evolution (15, 20, 21). The phylogenetic conservatism of the relative abundance of vertically transmitted microbes can also reinforce the resulting phylosymbiotic pattern (14). Alternatively, phylosymbiosis can emerge in the absence of vertical transmission, for instance, if the community assembly of microbes acquired from the environment is dominated by mechanisms of ecological filtering by the hosts and if the host traits involved in this filtering are phylogenetically conserved (12, 22). Some correlative approaches are available to investigate whether phylosymbiosis is supported by recent or ancient microbial divergences, such as the beta diversity sensitivity analysis (16, 23); however, they cannot directly assess whether a pattern of phylosymbiosis is linked to the vertical transmission of individual microbial lineages or to alternative processes, such as ecological host filtering.

An approach to inferring if and which microbial lineages are vertically transmitted among members of whole microbial communities is to use cophylogenetic methods, which quantify the congruence between trees (in terms of topology and relative times of divergence [19, 24]). When there is vertical transmission, transmitted microbial lineages follow host diversification, resulting in a congruence between the host phylogeny and that of each vertically transmitted microbial lineage (20). Conversely, in the absence of vertical transmission, such congruence may occur only in very specific cases, but it is not expected in general (12). However, testing for cophylogenetic signals is challenging for several reasons. First, transmitted microbes can experience not only vertical transmission but also events of horizontal switching between host lineages, which result in a loss of congruence between host and microbial phylogenies (25). Horizontal switches between particular host lineages are also expected to be more likely, for instance between closely related host species (that likely represent similar niches for the microbes) or between host species sharing the same geographic area (26). Such preferential host switches can strongly influence the observed cophylogenetic patterns (27). Second, because of the short length of the amplicons used to characterize microbiotas in high-throughput sequencing studies (e.g., 16S rRNA or internal transcribed spacer [ITS] genes [28]), the microbial DNA sequences available have accumulated only few mutations since their host diverged, providing limited information on the evolutionary history of the transmitted microbes (29). The difficulty in reconstructing a robust phylogeny for each microbial lineage is one of the most problematic challenges, especially as phylosymbiosis is often observed in recent host radiations, whereas it is "erased" by various factors such as diet shifts over longer timescales (30, 31). To address these challenges, we recently developed a likelihood-based model, called HOME (host-microbiota evolution) (32), designed to infer modes of microbial inheritance in the presence of host switches without reconstructing the phylogenetic tree of the microbial lineages. Instead, HOME directly models the evolution of the microbial DNA sequences on the host phylogeny, with potential

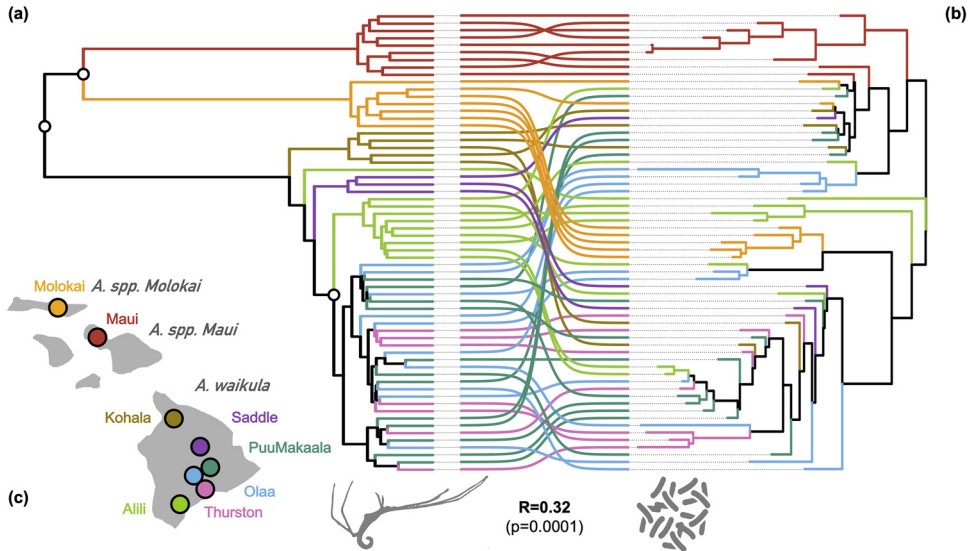

**FIG 1** Phylosymbiosis in the microbiota of Hawaiian spiders. (a) Phylogenetic tree of the host *Ariamnes* spiders obtained using ddRAD markers. Interspecific nodes are supported at 100%, whereas most intraspecific nodes have bootstrap supports greater than 70%. (b) Microbiota dendrogram obtained from the Bray-Curtis dissimilarities as a measure of beta diversities comparing the Swarm composition between spiders' microbiotas (similar patterns were obtained with 97% OTUs [data not shown]). On both trees, tips are colored according to the geographic area of the host spiders, and internal branches are also colored according to the area if all their descendant tips come from the same area; otherwise, they are in black. Colored links indicate which microbiotas belong to which spiders. The Mantel test between the host phylogenetic distances and the microbiota Bray-Curtis dissimilarities indicated a significant correlation ($R = 0.32$, $P = 0.0001$). In addition, white dots on the *Ariamnes* phylogenetic tree indicate nodes having a significant clade-specific phylosymbiosis (56). (c) Map of the Hawaiian archipelago where the *Ariamnes* spiders were sampled.

host switches, and tests whether this model of host-dependent evolution is supported in comparison with scenarios where microbial sequences evolve independently.

Here, we used HOME to investigate the modes of inheritance of the bacterial microbiota of a lineage of Hawaiian spiders in the genus *Ariamnes* that exhibit a significant pattern of phylosymbiosis (17). Our goal was to examine whether this empirical pattern of phylosymbiosis (at the whole-microbiota community level) is explained by vertically transmitted microbes (at the level of individual microbial lineages). The Hawaiian *Ariamnes* spiders are predators of other spiders (33) and are mostly restricted to wet forest habitats (34). Their radiation into 15 species within the last 2 million years across the Hawaiian archipelago shows a classic pattern of colonization from older to younger islands, which results in strong geographical clustering (34, 35). *Ariamnes* spiders are nonmodel organisms, and whether their associated microbes are vertically transmitted is unknown. In general, spider-associated bacteria can be either unspecific and transient (5) or intimate and maternally transmitted symbionts (36), as is commonly the case for endosymbiotic intracellular bacteria that colonize most spider tissues, including the midgut (17, 36). Previous work has shown that Hawaiian *Ariamnes* spiders have a diverse and relatively stable gut microbiota composed mainly of *Proteobacteria* and *Firmicutes*, and each spider is colonized by at least one endosymbiont from the genera *Wolbachia*, *Rickettsia*, and *Rickettsiella* (17). Because the different species of Hawaiian *Ariamnes* are highly specialized (similar habitat and very narrow and conserved diet relative to other spiders) and are closely related within an island, we thus expected (i) their niche conservatism to favor microbial vertical transmission, at least for the endosymbionts, and (ii) their geographical clustering to influence host switch dynamics.

## RESULTS

**Phylosymbiosis and inheritance of the microbiota of Hawaiian spiders.** Using previously published data on Hawaiian *Ariamnes* spiders and their associated microbiota (17), we confirmed that the evolutionary history of the 63 sampled *Ariamnes* spiders reconstructed using double digest restriction site-associated DNA (ddRAD) markers presented a clustering by geographic area (Fig. 1a). Looking at their associated microbiotas as a

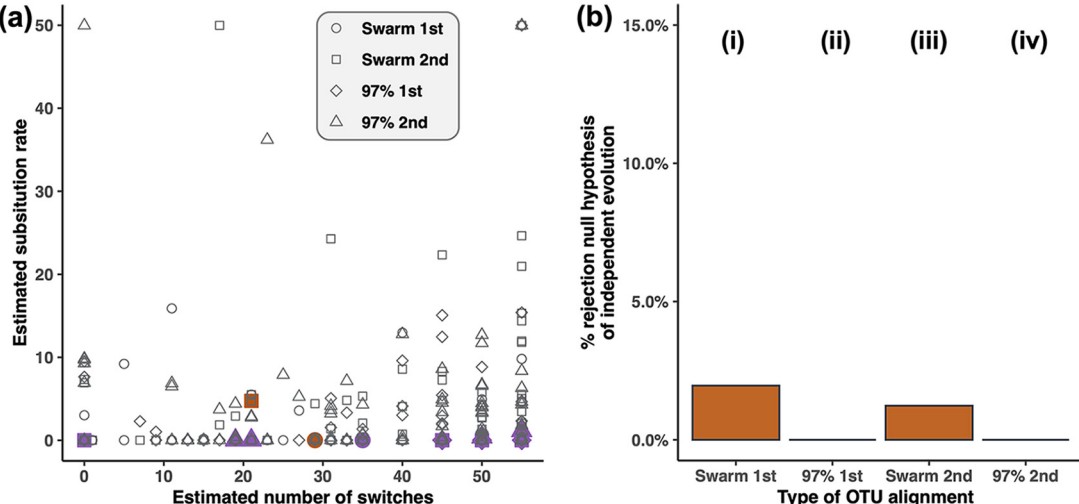

**FIG 2** HOME results on OTU alignments from the *Ariamnes* microbiota. (a) Estimated substitution rate as a function of the estimated number of switches for the different OTUs. Each point corresponds to an empirical OTU and its shape corresponds to the type of OTU (Swarm or 97% OTUs) and the representative sequence (the most [1st] or the second [2nd] most abundant sequence per host individual). OTUs corresponding to endosymbionts (*Wolbachia*, *Rickettsia*, and *Rickettsiella*) are in purple. The only two OTUs that rejected the null hypothesis of independent evolution are in orange. (b) Percentage of OTUs rejecting the null hypothesis of independent host-microbial evolution. The two OTUs that rejected the null hypothesis of independent evolution (columns i and iii) belong to the genera *Bacillus* and *Erythrobacter*, respectively, occurred in 12 and 51 host individuals, respectively, have 1 and 13 segregating sites, respectively, and have an estimated number of host switches of 12 and 21, respectively (very high compared to the number of hosts where they occurred): they are likely false positives.

whole using 16S rRNA amplicon sequencing, we found significantly congruent topologies between the host spider phylogeny and their microbiota dendrogram reconstructed via neighbor-joining from both weighted and unweighted beta diversities (Fig. 1b; also, see Fig. S1 at https://doi.org/10.17605/OSF.IO/5NJCQ). This phylosymbiotic pattern was also confirmed by Mantel tests, which indicated a positive association between host phylogenetic distances and microbiota dissimilarities, although it was significant only when weighted beta diversity indices were used (see Fig. S1 at https://doi.org/10.17605/OSF.IO/5NJCQ). We also measured significant phylosymbiosis in some subclades of the spider phylogeny (Fig. 1a), not only at the level of the whole *Ariamnes* clade. Phylosymbiosis was detected for both operational taxonomic unit (OTU) clusterings we performed (Swarm OTUs and OTUs at 97%) (see Materials and Methods; also, see Fig. S1 at https://doi.org/10.17605/OSF.IO/5NJCQ).

Next, we used HOME to infer the inheritance modes for each of the OTUs that were shared by at least 5 spider individuals (we refer to these OTUs as the "shared OTUs," in contrast to "unshared OTUs," which were detected in fewer than 5 spider individuals). Shared OTUs represent 96 Swarm OTUs (out of 413 Swarm OTUs, i.e., 23% of the total Swarm OTUs) and 103 OTUs at 97% (out of 414 OTUs at 97%, i.e., 27% of the total OTUs at 97%). These shared OTUs presented a relatively low number of segregating sites (see Fig. S2 at https://doi.org/10.17605/OSF.IO/5NJCQ) and occurred in 5 to 55 host individuals (see Fig. S3 at https://doi.org/10.17605/OSF.IO/5NJCQ), but they represented 88% (Swarm clustering) and 89% (clustering at 97%) of the total bacterial reads. Given that HOME uses intra-OTU variation to detect vertically transmitted OTUs, HOME can be run only on the OTU alignments having at least one segregating site. When selecting the most abundant sequence per host individual and assembling the OTU alignment across all host individuals, only 51 Swarm OTUs and 66 OTUs at 97% had at least one segregating site in the OTU alignment, while we had 81 Swarm OTUs and 90 OTUs at 97% when selecting the second most abundant sequence. When HOME was applied to these OTUs, no OTU, including the endosymbiotic ones, rejected the null hypothesis of host-independent evolution (Fig. 2), except 2 of the 132 (51 + 81) Swarm OTUs tested. This ratio (2/132 [1.5%]) falls into the global type I error of HOME (i.e., the percentage of environmentally acquired OTUs that are incorrectly inferred as being vertically transmitted by the model [32] [see Fig. S4b at https://doi.org/10.17605/OSF.IO/5NJCQ]), so these 2 OTUs are likely false positives. In addition, we confirmed

that these small numbers of shared OTUs were mainly responsible for the global pattern of phylosymbiosis. Indeed, we still found a significant pattern of phylosymbiosis when randomizing the unshared OTUs while keeping the shared OTUs untouched, but conversely, the Mantel correlations were no longer significant when randomizing the shared OTUs and keeping untouched the unshared ones (see Table S1 at https://doi.org/10.17605/OSF.IO/5NJCQ).

**Testing the performance of HOME with simulations.** The nondetection of vertically transmitted OTUs could come from the slow evolution of the 16S marker gene, which has accumulated very few nucleotide substitutions during the *Ariamnes* spider radiation, in particular on the short (310-bp) V1-V2 region. Thus, to test the effect of low substitution rates ($\mu$) on the performance of HOME, we simulated OTU alignments with very low numbers of segregating sites ($\mu = 0.1$) (see Materials and Methods; also, see Fig. S5 at https://doi.org/10.17605/OSF.IO/5NJCQ) similar to those of the empirical OTU alignments (see Fig. S2 at https://doi.org/10.17605/OSF.IO/5NJCQ). Compared to more variable alignments ($\mu = 1.5$) (see Fig. S5 at https://doi.org/10.17605/OSF.IO/5NJCQ), we found as expected that the ability to recover simulated parameters (i.e., the number of host switches [$\xi$] and the substitution rate [$\mu$]) decrease when the simulated substitution rate is low (see Fig. S6 and S7 at https://doi.org/10.17605/OSF.IO/5NJCQ). The approach tends to underestimate the inferred number of host switches when there are many ($F_{1,68}=11.7$, $P = 0.001$) (see Fig. S8a at https://doi.org/10.17605/OSF.IO/5NJCQ); however, we still found a positive correlation between the number of simulated ($\xi$) and estimated ($\hat{\xi}$) host switches. Similarly, HOME correctly estimates the simulated low substitution rate but tends to overestimate it when $\xi$ is large ($t_{69}=3.0$, $P = 0.004$) (see Fig. S8c at https://doi.org/10.17605/OSF.IO/5NJCQ). $\hat{\xi}$ and $\hat{\mu}$ values estimated from OTU alignments simulated independently from the host phylogeny were significantly higher than those estimated from vertically transmitted OTUs (see Fig. S8 at https://doi.org/10.17605/OSF.IO/5NJCQ), and the null hypothesis of host-independent evolution was never rejected for these alignments (Fig. 3; also, see Fig. S8d at https://doi.org/10.17605/OSF.IO/5NJCQ). Conversely, the null hypothesis of host-independent evolution was on average rejected for 50% of the OTU alignments vertically transmitted with no or few (fewer than 15) host switches (intermediate statistical power) (Fig. 3; also, see Fig. S8d at https://doi.org/10.17605/OSF.IO/5NJCQ); this statistical power decreased with the number of simulated host switches (Fig. 3; also, see Fig. S8d at https://doi.org/10.17605/OSF.IO/5NJCQ). These results also depend on the number of hosts in which the OTU is found, and the statistical power decreased at 40% for OTUs occurring in only 20 hosts and below 10% for OTUs occurring in only 5 hosts (see Fig. S9 at https://doi.org/10.17605/OSF.IO/5NJCQ). Given the statistical power of HOME in this system (50% when the number of segregating sites is low and the number of hosts in which the OTU is found is high), we can conclude that, if any, there are at most 4 core OTUs that are vertically transmitted (see Fig. S4a at https://doi.org/10.17605/OSF.IO/5NJCQ) and no more than 6 OTUs occurring in only 20 hosts that are vertically transmitted. Therefore, most bacterial OTUs from the *Ariamnes* microbiota seem to evolve independently from their host spiders.

Similarly, we tested whether preferential host switches occurring in the Hawaiian archipelago could affect the performances of HOME, and we found that we were able to recover simulated parameter values (see Fig. S6 and S7 at https://doi.org/10.17605/OSF.IO/5NJCQ), especially when the simulated substitution rate was high ($\mu = 1.5$). Both types of preferential host switches (phylogenetic relatedness or geographic dependencies; see Materials and Methods) affected the estimation of the parameters in the same way: the estimated number of switches ($\hat{\xi}$) and the estimated substitution rates ($\hat{\mu}$) tend to decrease when the effect of preferential host switches is higher (see Fig. S6 and S7 at https://doi.org/10.17605/OSF.IO/5NJCQ). Transmitted OTUs simulated under preferential host switches tend to be inferred as strictly vertically transmitted OTUs (i.e., they tend to have estimated parameters similar to those of OTUs simulated under strict vertical transmission: small $\hat{\xi}$ and $\hat{\mu}$. Similarly, the statistical power of the test of host-independent evolution significantly increased with preferential host switches (Fig. 3), whatever the type of preferential host switches or the simulated substitution rate: OTUs simulated with a vertical transmission and preferential host

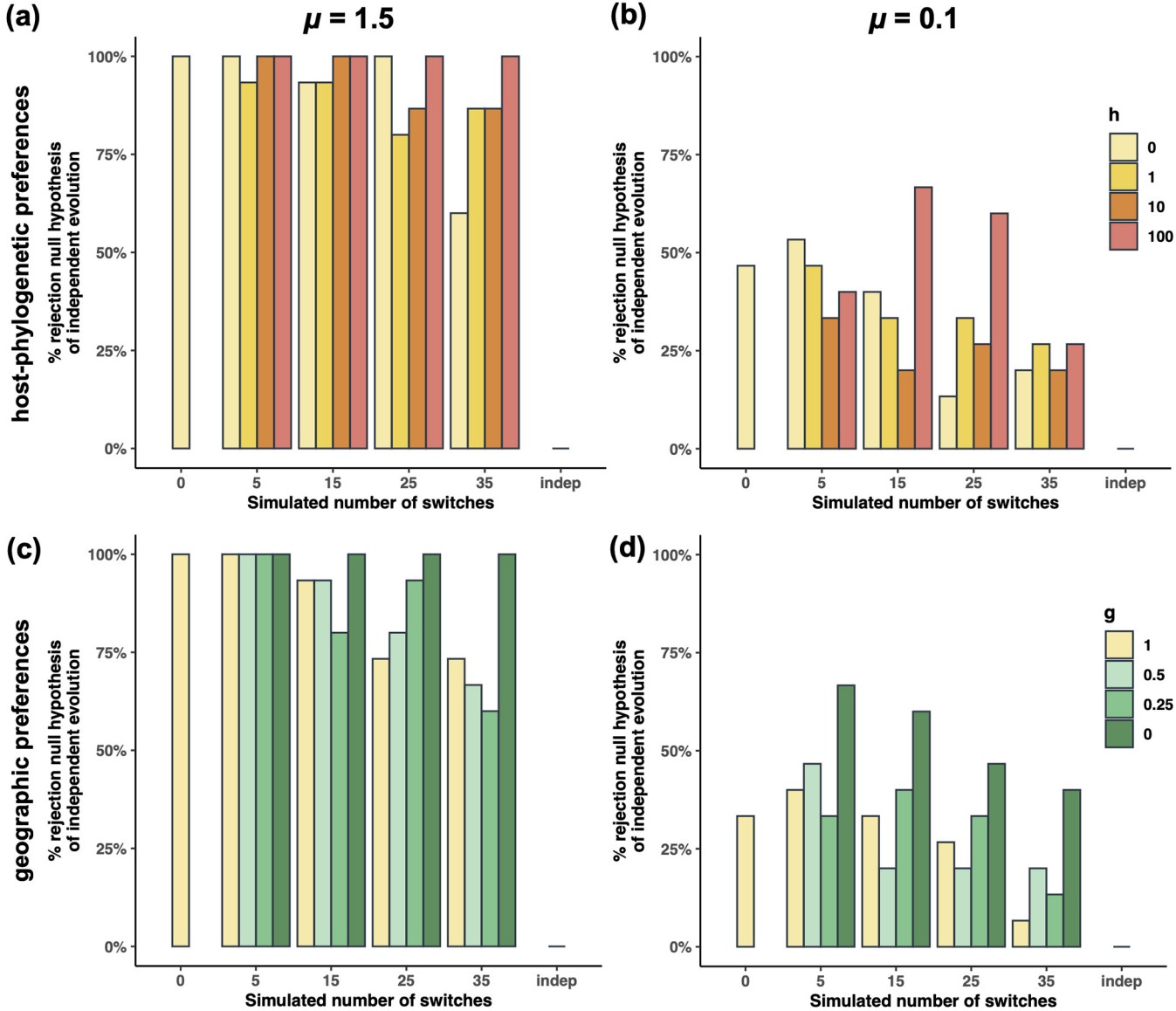

**FIG 3** Evaluating the effect of substitution rates and preferential host switches on the rejection of the null hypothesis of host-independent evolution. Percentage of simulated OTU alignments that rejected the null hypothesis of independent host-microbial evolution, according to the different evolutionary scenarios simulated on the 63-tip *Ariamnes* tree: strict vertical transmission ($\xi = 0$), vertical transmission with $\xi$ host switches ($\xi > 0$), or independent evolution. Four cases were tested: high substitution rate (a and c) relative to low substitution rate (b and d) and the possibility of host-phylogenetic preferences in the simulated switches (a and b) or geographic preferences (c and d). Simulated values $h = 0$ and $g = 1$ correspond to uniformly distributed host switches, whereas the values $h > 0$ and $g < 1$ indicate a certain degree of preferential host switches.

switches were more frequently inferred as vertically transmitted than those that experienced uniformly distributed host switches (Fig. 3). Thus, preferential host switching is very unlikely to negatively affect the ability of HOME to detect vertically transmitted OTUs.

## DISCUSSION

We used HOME, a cophylogenetic model-based approach well adapted to microbiota data sets, to assess whether phylosymbiosis can emerge without vertical microbial transmission in an empirical system (14). We found that the significant phylosymbiosis pattern of microbiotas of recently diverging host spiders across the Hawaiian archipelago is likely not explained by vertically transmitted microbes. Using simulations, we showed that this result is not due to the low number of segregating sites or to preferential host switches.

**Inferring vertically transmitted symbionts.** Simulations with very low substitution rates showed that the statistical behavior of HOME remains acceptable when there are

very few segregating sites in the microbial alignment: although the power to reject the null hypothesis of host-independent evolution is reduced, the type I error rate remains low (i.e., symbionts that evolved independently from the host phylogeny are rarely inferred as vertically transmitted). This indicates that HOME can be applied to molecular microbial markers that evolve slowly (e.g., 16S rRNA gene) and when host divergences are recent, such as among the species of *Ariamnes* spiders included here, which diverged only <2 million years ago (34). The ability to study the mode of inheritance in microbiota over short timescales is one of the main advantages of HOME compared to other models, such as Jane (37) and ALE (38), that need to reconstruct the microbial phylogenetic trees.

If a transmitted OTU is found in only a limited number of hosts, either because it is not present or because it is not sampled due to detection limitation, HOME will suffer from a low statistical power, meaning that it will fail to reject the null hypothesis of independent evolution. A high sequencing coverage is therefore required to minimize undersampling and maximize the detection of transmitted OTUs.

The presence of preferential host switches does not reduce the statistical power of HOME in comparison with its power in the presence of uniformly distributed host switches; on the contrary, microbial alignments simulated under preferential (versus uniformly distributed) host switches are more likely to be inferred as strictly vertically transmitted symbionts. This result is not surprising, as, if host switches preferentially occurred between host lineages that are phylogenetically related, the microbial phylogeny tends to be more congruent with the host phylogeny than if host switches occurred uniformly. Similarly, as phylogenetically related host lineages also tend here to occupy the same geographic area, we could expect the microbial phylogenies resulting from geography-dependent host switches to be more similar to the host phylogeny than if host switches occurred uniformly. Thus, in our simulated scenarios, preferential host switches strengthen the host phylogenetic signal within the microbial alignment and increase the ability of HOME to detect vertically transmitted OTUs.

Our test of the ability of HOME to detect preferential host switches if they occurred on the *Ariamnes* phylogeny showed that such detection is difficult and that both phylogenetically and geographically driven host switches leave similar signals in the microbial alignments and are therefore undifferentiable (see supplemental results 1 at https://doi.org/10.17605/OSF.IO/5NJCQ). This does not imply that preferential host switches cannot be inferred from any host phylogeny, as the *Ariamnes* phylogeny has a strong geographic structure that renders this inference particularly challenging. Simulations on other hosts phylogenies are required to provide definitive conclusions about these possibilities, as such detection would be informative regarding the host switching processes (24).

**Absence of vertically transmitted microbes in *Ariamnes* spiders.** The microbiota of the *Ariamnes* spiders showed a low proportion of shared OTUs, and given that the sequencing coverage was high (low risk of detection limitation), it suggests that microbiota composition across spiders is quite variable. By applying HOME, we showed that these bacterial OTUs did not reject the null hypothesis of host-independent evolution. Given the statistical power of HOME when the number of segregating sites is low, we can conclude that there are likely fewer than five vertically transmitted core OTUs in this system. The two OTUs that rejected the null hypothesis of independent evolution with HOME, which respectively belong to the ubiquitous genera *Bacillus* and *Erythrobacter*, have high estimated numbers of host switches, which likely resulted in incongruent cophylogenetic patterns (Fig. 2). In addition, their estimated parameters fit into the distribution of parameters obtained for the nontransmitted OTUs (Fig. 2). Thus, we can conclude that these OTUs are likely false positives. Instead, most bacteria are probably acquired in the environment at each generation by spider individuals. Although the ability of HOME to infer vertically transmitted OTUs occurring in only few hosts is limited, the fact that these OTUs are absent in most of the host individuals suggests that these OTUs are facultative symbionts, rather than specific vertically transmitted symbionts. The spider microbiota assembly is thus likely not determined by the vertical inheritance of microbial lineages in this system. Such results were partially expected given that spider microbiotas can show a very high heterogeneity and that feeding experiments have recently demonstrated the lability of the microbiota composition

according to the spider's diet (5). This corroborates the fact that the degree of conservatism and the functional relevance of the microbiota are highly variable across the animal kingdom, especially within arthropods in which the microbiota composition ranges from mainly transient microbes acquired from the environment (39, 40) to striking examples of vertically transmitted microbes (8).

One could argue that not detecting vertically transmitted bacterial OTUs in this study comes from the fact that the 16S rRNA marker evolves too slowly to have accumulated any segregating sites in the nucleotide alignment of vertically transmitted OTUs. Nucleotide alignments without segregating sites could thus correspond to vertically transmitted OTUs. However, the nucleotide substitution rate of the 16S rRNA gene is estimated to be 1% per 50 million years in bacteria (29). Given that the sum of the branch lengths of the phylogenetic tree of the *Ariamnes* spiders represents a total of 13 million years of nucleotide evolution, we expect on average at least one segregating site per 300-bp alignment of vertically transmitted bacteria. Given that substitution rates of symbiotic bacteria are higher because of their small population size compared to free-living bacteria (8, 41), this would confer even more variability within the OTU alignments of vertically transmitted bacteria. Therefore, it is unlikely that there are vertically transmitted microbes among *Ariamnes* microbiota but that they do not have segregating sites. Future works specifically targeting microbial marker genes or another 16S rRNA region that are longer or evolve faster would help to confirm the absence of vertically transmitted microbes among *Ariamnes* spiders. Furthermore, future studies targeting other arthropod clades would allow to test the generality of our results in arthropods.

**Other drivers of phylosymbiosis.** Our study highlights an empirical system in which phylosymbiosis is likely explained by processes other than vertical transmission (19). First, phylosymbiosis can emerge through the existence of a simple ecological filtering during host colonization, as has been hypothesized before (12) and demonstrated using simulations (22). Indeed, if the microbiota is entirely acquired from the environment and if its assembly is influenced by host traits (e.g., gut pH or diet) that are phylogenetically conserved, then the microbiota composition will be more similar between closely related than distantly related species. Such mechanisms could act in the microbiota assembly of *Ariamnes* spiders and be responsible for the observed pattern of phylosymbiosis. Experimental studies and/or a better characterization of host traits would be required to confirm that microbiota composition is primarily driven by phylogenetically conserved host filtering mechanisms. Second, phylosymbiosis can emerge if phylogenetically related host lineages tend to occupy similar geographic areas and there is a geographic structure in the environmental pools of available microbes (19). In *Ariamnes* spiders for example, phylogenetically related lineages tend to occupy the same island, and each island is characterized by one dominant endosymbiont (*Wolbachia*, *Rickettsia*, and *Rickettsiella* [17]). They are well known to be transmitted from generation to generation through direct transfer in the maternal germ line (8), but our analyses suggest that they are not vertically transmitted over long timescales (Fig. 2). Instead, this suggests that the temporal turnover of the endosymbionts is relatively high compared to the timescale of host diversification and that their epidemic spread is influenced by island structure. In arthropods, endosymbionts can be either conserved over long timescales and codiversify with their hosts (42) or frequently horizontally transmitted between host lineages (43). Which processes contribute to this variability remains unclear. We suspect that in *Ariamnes* spiders, predation on other arthropods and cannibalism can increase the likelihood of endosymbiont horizontal transmissions over long timescales (17, 44). Altogether, this suggests that host diversification and microbiota evolution can happen at two decoupled timescales, even in the presence of phylosymbiosis.

Our results in *Ariamnes* spiders do not imply that phylosymbiosis in other host-microbiota systems is not (at least partly) explained by vertical transmission. For instance, bacterial vertical transmission occurs in the gut microbiota of mammals (23, 30, 32, 45), where it generates stronger phylosymbiosis than host filtering alone (22). Importantly, phylosymbiosis indicates a degree of host-phylogenetic conservatism in the many processes involved in microbiota assembly during host evolution (46) but does not by itself inform on the nature

of these processes. Model-based approaches such as HOME can provide a more precise characterization of these nonexclusive processes, and more work in this direction is needed to improve our understanding of microbiota assembly and evolution.

## MATERIALS AND METHODS

**Study system: microbiotas of Hawaiian spiders.** We used previously published data on Hawaiian *Ariamnes* spiders and their associated microbiota (17). One hundred twenty-three Hawaiian *Ariamnes* spider hosts were sampled on 3 islands under the leaves in understory vegetation at a total of 8 sites within the wet forest (similar temperature and precipitation), with the goal of minimizing environmental variation and to guarantee that all individuals had similar microniches (Fig. 1) (17). Genome-wide sequencing of the host spiders and 16S rRNA metabarcoding of their associated microbiota were performed for 63 individuals (34) from one species (*Ariamnes* sp. nov.) on Molokai, one (*Ariamnes melekalikimaka*) on West Maui, and 6 populations of one species (*Ariamnes waikula*) on Hawaii Island. Our study concerns these 63 individuals.

We used the robust phylogenetic tree of the 63 host individuals reconstructed using genomic ddRAD markers published in reference 17. In short, the phylogenetic tree was obtained using the Stacks pipelines (47) and IQ-TREE (48) with 100 bootstraps (see supplemental methods 1 at https://doi.org/10.17605/OSF.IO/5NJCQ and reference 17 for details). The tree was made ultrametric using r8s (49) but not calibrated in absolute time, which was not necessary for our analyses. We also used the short DNA metabarcoding sequences from the 16S rRNA gene generated in reference 17 to characterize the spider-associated bacterial communities. Microbiota DNA extractions were performed using the Gentra Puregene tissue kit (Qiagen, Hilden, Germany) on the spiders' abdomens, which contain the gut as well as other organs, such as the gonads (5). Bacterial 16S rRNA genes were targeted using a primer pair amplifying approximately 310 bp of the V1-V2 variable regions (50). The amplicon library was sequenced using Illumina MiSeq technology generating $2 \times 300$-bp paired reads. Negative controls (extraction blanks and no-template controls) were carefully performed.

The corresponding microbiota raw data (obtained from reference 17) encompassed 4,932,236 microbial reads, which were assembled, demultiplexed, and quality checked using VSEARCH v2 (51). We clustered reads according to their sequence similarities into operational taxonomic units (OTUs) using two different algorithms. We first used Swarm v2 (52), with the fastidious option, which groups reads into OTUs without specifying a global similarity threshold and thus can accurately identify clustered structures at a finer scale. We performed a second clustering using a classical OTU clustering at 97% similarity with VSEARCH. We assigned a taxonomy to each OTU using the SILVA database (53), filtered out chimeras, and built an OTU table indicating for each OTU its abundances in the different spiders' microbiotas. Finally, nonbacterial OTUs and contaminant OTUs present in high abundances in the negative controls were filtered out of the OTU table. We obtained a total of 413 Swarm OTUs and 414 OTUs at 97%, which correspond to totals of 1,297,307 and 1,178,325 reads, respectively. Rarefaction analyses confirmed that the sequencing coverage was sufficient to get most of the bacterial diversity in the majority of the samples (17).

**Assessing phylosymbiosis.** We assessed phylosymbiosis using two different tests (13, 14): (i) a Mantel test between *Ariamnes* phylogenetic distances and microbiota beta diversities and (ii) a matching cluster analysis (54), which tested the topological congruence between the *Ariamnes* phylogeny and microbiota dendrograms constructed from microbiota beta diversities (see supplemental methods 2 at https://doi.org/10.17605/OSF.IO/5NJCQ). For each test, we used both weighted (i.e., accounting for abundance) and unweighted (i.e., presence-absence) beta diversity metrics, computed using OTU tables rarefied at 5,000 reads per sample. For weighted metrics we also used relative abundances instead of rarefactions (55). Additionally, we used clade-specific Mantel tests to investigate whether phylosymbiosis was also present in some subclades of the *Ariamnes* phylogeny, as described in reference 56. All phylosymbiosis analyses were performed on both clusterings (Swarm OTUs and OTUs at 97%). Similar trends using amplicon sequence variants (ASVs) instead of OTUs were reported in reference 17, so we did not replicate the analyses using ASVs here.

**Inferring vertically transmitted symbionts.** The inheritance modes among the microbial OTUs were inferred independently for each OTU using HOME (32). Each OTU was characterized by a nucleotide alignment made of the microbial sequences obtained across the different host lineages, with at most one single representative DNA sequence per host individual. In short, HOME uses the intra-OTU variation (segregating sites) contained in the nucleotide alignment to test whether each microbial OTU has likely evolved on the host phylogeny by vertical transmission or alternatively has been acquired from the environment. It assumes that for a given OTU, microbial populations, represented by a DNA sequence in each host lineage, (i) are vertically transmitted along branches on the host phylogeny; (ii) can experience DNA substitutions with a rate $\mu$ ($\mu$ is constant for a given OTU but can vary across OTUs), which generates segregating sites in the OTU alignment; (iii) are inherited at host splitting events by the two daughter host lineages; and (iv) can experience a certain number of host switches ($\xi$), where one microbial sequence from a donor host branch is horizontally transmitted at a given time to a receiving branch, where it replaces the previous sequence. By default, host switches are assumed to be uniformly distributed on the host branches. For each OTU independently, HOME uses a combination of likelihood-based and simulation-based approaches to estimate $\xi$ and $\mu$ and to test whether a scenario of transmission is more likely than a scenario of host-independent evolution where the links between microbial sequences and host lineages are randomized (see reference 32 for more details). Importantly, because HOME uses the intra-OTU variation, HOME cannot be run on ASVs that are unique sequences (obtained after removing sequencing/PCR errors) and therefore present no "within-unit" variation. HOME was therefore run only on Swarm OTUs and OTUs at 97%.

To run HOME, we selected the OTUs shared by at least five individual spiders, as HOME cannot perform well with lower occurrence. In addition, based on the content of negative PCR controls and on previous estimates (57), we assumed that if an OTU occurred with fewer than 5 reads in a spider, it was likely the result of cross-contaminations during the library preparation, and we considered the OTU absent in this spider's

microbiota. For a given OTU, we selected one representative sequence per host spider by taking the most abundant read confidently assigned to this OTU present in the spider microbiota. Neglecting the microbial intraindividual variation is equivalent to considering that host individuals are colonized by only one unique microbial strain per OTU and that the intraindividual variability observed in the data is caused by PCR and sequencing errors. To relax this hypothesis, we repeated the analyses by instead picking, when available, the second most abundant read as the representative sequence of one OTU in one host spider, although these sequences were likely artifacts (58). We then assembled the sequence alignment for each OTU and ran HOME separately. As HOME models neither microbial extinction nor the sampling process (i.e., the probability of sampling a given OTU if present), when an OTU was absent from a given host spider, which can occur either because the OTU is truly absent or because it is not detected, we pruned this host out of the spider phylogenetic tree (32).

Finally, to confirm that these few OTUs shared by multiple host individuals contributed to the pattern of phylosymbiosis at the whole-community level (encompassing all shared and unshared OTUs), we randomized the unshared OTUs among *Ariamnes* samples while keeping the shared OTUs untouched (and vice versa) and reassessed phylosymbiosis using Mantel tests.

**Simulating the performance of HOME under low intra-OTU variation and preferential host switches.** To confirm that HOME would detect vertically transmitted OTUs in *Ariamnes* microbiotas despite the recent host divergences and the likely occurrence of preferential host switches, we tested the performance of HOME using simulations by (i) checking the statistical power of HOME when there are only few segregating sites in the microbial OTU alignments and (ii) testing the effect of phylogenetically driven and geographically driven host switches.

First, we simulated, on the *Ariamnes* host tree, microbial phylogenies of OTUs evolving under vertical transmissions with 0, 5, 15, 25, or 35 host switches or OTU phylogenies that evolved independently from the host phylogeny. For each scenario, we simulated 15 independent OTUs. Given the phylogenetic tree of each OTU, we simulated the corresponding evolution of a nucleotide alignment of 300 bp, with a probability 0.1 for each site to be variable under a stochastic K80 (59) nucleotide substitution process (with a ratio of transition/transversion rate $\kappa$ of 0.66) and we tested the effect of a very low relative substitution rate ($\mu = 0.1$), compared to intermediate values ($\mu = 1.5$). We then ran HOME on each alignment independently. Furthermore, as some OTUs occurred in only few host individuals (because they were either absent or undetected), we tested the effect of low OTU occurrence on the statistical power of HOME by simulating the alignment corresponding to each OTU on a host phylogeny randomly pruned to 5 or 20 tips.

Next, we simulated vertically transmitted microbial symbionts which experience events of host switches driven by host relatedness (27). We considered that the probability that there is a host switch between times $t$ and $t+dt$ depends on the phylogenetic relatedness between pairs of host lineages among the $N(t)$ other coexisting hosts at this time, such that:

$$\mathrm{P}(\text{host switch at time } t) \approx \frac{1}{N(t)-1} \sum_{i \in [1, N(t)]} \sum_{j \in [1, N(t)]; j \neq i} e^{-h d_{i,j}(t)}$$

where $d_{i,j}(t)$ represents the phylogenetic distance, measured as branch length, between the coexisting hosts $i$ and $j$ at time $t$ and $h$ is a parameter tuning the effect of host phylogenetic relatedness (if $h$ is 0, there is no effect of host relatedness, and the higher $h$ is, the more likely it is that host switches occur between closely related hosts). If a host switch occurs at time $t$, a pair $(i,j)$ of host lineages involved in the host switch is chosen proportionally to $e^{-h d_{i,j}(t)}$ (if $h$ is greater than 0, pairs of hosts that are phylogenetically distant [i.e., large values for $d_{i,j}(t)$] are unlikely to be chosen). For a $h$ value of 0 (uniform distribution of host switches), the probability of a switch is proportional to the number of host lineages at time $t$, and pairs of hosts involved in the switch are chosen uniformly. We performed the same simulations as detailed above (no host switch) using 4 $h$ values: $h \in (0, 1, 10, 100)$.

Similarly, we simulated vertically transmitted microbial symbionts which experience events of host switches that are more likely between hosts sharing the same geographical area (i.e., same sampling site). We assumed that hosts occupy a unique discrete area, which can change through time. We first reconstructed the ancestral biogeography of the host phylogeny using stochastic mapping (make.simmap function, phytools R package [60]), considering that migrations between areas are punctuated events occurring with different estimated probabilities, represented by a symmetrical matrix of transition ($Q$). We considered $g$, the probability for a host switch to occur between hosts from different areas divided by the probability for a host switch to occur between hosts of the same area: if $g$ is equal to 1, host switches between hosts from different areas are as likely as host switches within the same area, whereas a $g$ value of 0 corresponds to a scenario where host switches occur only between hosts sharing the same area. We considered that the probability that a host switch occurs at time $t$ depends on the total number of hosts sharing the same area at time $t$ and the number of hosts alone on their own area at time $t$ multiplied by $g$, such that:

$$P(\text{host switch at time } t) \sim \sum_{A \in \text{areas}, \, N_A(t) > 1} N_A(t) + \sum_{A \in \text{areas}, \, N_A(t) = 1} g \, N_A(t)$$

with $N_A(t)$ being the number of hosts in area $A$ at time $t$.

If a host switch occurs at time $t$, pairs of hosts are then chosen with a relative weight of 1 if they are in the same area, or $g$ if not. These simulations could be further improved by considering geographic distances between the different areas. We performed the same simulations as detailed above using 4 $g$

values: $g \in$ (1, 0.5, 0.25, 0). For each simulated OTU, we used a different stochastic mapping of the ancestral biogeography to consider the uncertainty in the reconstruction.

In addition to these tests, we investigated the ability of HOME to detect preferential host switches on the *Ariamnes* tree (see supplemental methods 3 at https://doi.org/10.17605/OSF.IO/5NJCQ).

**Data availability.** The raw data can be found on dryad (https://doi.org/10.5061/dryad.nzs7h44qj). The implementation of HOME used in this study and a tutorial are available on GitHub (https://github .com/BPerezLamarque/HOME).

## ACKNOWLEDGMENTS

We acknowledge E. Armstrong, J. Lim, A. Rueda, T. Schol, S. Kennedy, and S. Prost for helpful discussions. B.P.-L. and H.M. also thank D. de Vienne, M. Elias, M.-A. Selosse, F. Martos, F. Delsuc, M. Roy, L. Aristide, C. Fruciano, I. Quintero, S. Lambert, I. Overcast, O. Maliet, A. Silva, and G. Sommeria for comments on an early version of the manuscript. We also thank the anonymous reviewers for their constructive comments.

This work was supported by a doctoral fellowship from the École Normale Supérieure de Paris to B.P.L. and the École Doctorale FIRE – Program Bettencourt. R.G.G. acknowledges support from the National Science Foundation, DEB 1241253. H.M. acknowledges support from the European Research Council (grant CoG-PANDA). B.P.-L. and H.M. also acknowledge support from the iBioGen Twinning project (H2020 research and innovation program, grant agreement 810729).

All authors designed the study. B.P.-L., H.K., and R.G.G. gathered the data, and B.P.-L. performed the analyses. B.P.-L. and H.M. wrote the first version of the manuscript, and all authors contributed to the revisions.

We declare that we have no conflict of interest.

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
