## [Reviewer comments · mSystems]

Limited evidence for microbial transmission in the phylosymbiosis between Hawaiian spiders and their microbiota

Benoit Perez-Lamarque, Henrik Krehenwinkel, Rosemary Gillespie, and H el ene Morlon

Corresponding Author(s): Benoit Perez-Lamarque, IBENS

Review Timeline:

Submission Date:	September 3, 2021
Editorial Decision:	October 18, 2021
Revision Received:	December 16, 2021
Accepted:	December 16, 2021

Editor: Sarah Hird

Reviewer(s): The reviewers have opted to remain anonymous.

Transaction Report:

DOI: <https://doi.org/10.1128/mSystems.01104-21>

October 18, 2021

Dr. Benoit Perez-Lamarque
IBENS
Paris
France

Re: mSystems01104-21 (Limited evidence for microbial transmission in the phyllosymbiosis between Hawaiian spiders and their microbiota)

Dear Dr. Benoit Perez-Lamarque:

Thank you for submitting your manuscript to mSystems. We have completed our review and I am pleased to inform you that, in principle, we expect to accept it for publication in mSystems. However, acceptance will not be final until you have adequately addressed the reviewer comments.

Editor Notes (from Sarah):

This is a very nice manuscript and I look forward to seeing it in press. In addition to addressing the review comments, I would like to see the following addressed:

- (1) Fig. 1 - should the internal branches of (A) be colored as they are or black as in (B)? Also, could you please mark which nodes on the trees support phyllosymbiosis? I think it's helpful to readers to put a circle or star on the congruent nodes (so they do not have to search for them).
- (2) I agree with reviewer 1's point about the colors in fig. 2a being too similar; I recommend changing the colors and possibly adding a shape to make it easier to see (instead of all points being circles).
- (3) I also had a question similar to reviewer 1 about the justification of using OTUs over ASVs.

Preparing Revision Guidelines

Sincerely,

Sarah Hird

Editor, mSystems

Journals Department

Reviewer comments:

Reviewer #1 (Comments for the Author):

Thank you for the opportunity to review this paper, I thoroughly enjoyed reading it. The authors address an interesting question regarding the patterns that regulate phylosymbiosis and take a novel approach to disentangle whether vertical transmission or other processes may be responsible for these patterns. The authors laid out their predictions clearly, and I felt that they did a good job explaining the function of their model in the context of their data. The development of a new computational tool to address questions regarding microbial-host coevolution and phylosymbiosis is likely to be of interest to many readers.

In general, I have a few comments I would like the authors to address:

- 1.) What is the justification for using OTUs instead of ASVs in your analyses? I appreciate the comparison of using swarm and 97% OTU clustering, but I'm interested in why you avoided using ASVs.
- 2.) While I understand that the focal interest of this research team is the *Ariamnes* species complex in Hawaii, I'm wondering if it wouldn't bolster the paper to conduct a similar test of HOME using information from additional lineages of spiders or arthropods. There are a number of difficulties your system presents for your model, as you mentioned throughout the manuscript, and I think it would be useful to see another case study. If not that, a further explanation of why things like endosymbionts might not appear to be vertically transmitted here. What about these spiders might explain this pattern?
- 3.) Also, given the geographic and phylogenetic structure of your study system, how can you confirm the validity of the model's ability to differentiate between phylogenetically- and geographically-driven host switches?
- 4.) While I liked the visualizations of the data, they could be improved by using a colorblind friendlier palette. In particular, the colors used for Molokai and Kohala in Figure one are very similar, and the colors used in Figure 2a are quite difficult to differentiate.

Reviewer #2 (Comments for the Author):

This manuscript uses sequencing data from *Ariamnes* spiders and its microbiota - displaying phylosymbiosis pattern - to present an tool (HOME) developed by them that solve the problem of distinguish symbionts vertically transmitted from the environmentally acquired - since both can lead to this pattern. In addition to presenting an example of phylosymbiosis pattern caused by non-vertically transmitted symbionts, the authors perform simulations that confirm the resultus. They show the HOME model is sensible enough to detect vertical transmission even with a low number of segregating sites - as may be expected in 16S rRNA data normally used in microbiome studies. I think this manuscript is important for two main reasons. First, for confirming that patterns of phylosymbiosis can be caused by symbionts maintained in populations in different ways (vertically transmitted or environmentally acquired). Second, because once a phylosymbiosis pattern is observed - what seems to be quite prevalent in several evolutionary lineages - one obvious next question is how these symbionts are being maintained/acquired. Selection will act on microbially influenced host phenotypes that are heritable, and it can be directly on the symbiont or on a host trait that influences the environmental acquisition.

The authors were able to take advantage of an interesting dataset for their goal, the paper is well presented, concise and to the point. This manuscript will be of interest to those working directly with phylosymbiosis hypotheses, but also to the great mass of researchers currently working with microbiome data - as for those working with non-model organisms for which is more complicated to experimentally test modes of symbiont transmission.

I do not have any major concerns with the paper but have several minor recommendations for improving the clarity of the paper.

(Line 26) Include the word "vertically" when talking about vertical transmission.

(Line 48) It is the host that filters, and that is why we observe phylosymbiotic pattern. Thus, I would suggest changing "including environmental filtering" to "including host-filtering environmental acquired microbes".

(Line 49) "But which processes drive phylosymbiosis in nature remains unknown" you mentioned can be both, right? So it seems the problem is distinguishing it. I would suggest: "But distinguishing the processes driving phylosymbiosis in nature remains challenging".

(Line 88) For me it is not very clear why tree congruence is not expected when symbionts are environmentally acquired on a dependence on host-filtering. Also, the cited reference (12) actually shows that even when symbionts are acquired from the environment it can lead to mirrored phylogenies, due to host filtering. You mentioned this in discussion as well (Line 314).

(Line 129) These Ariamnes spiders are non-model organisms and whether or not their associated microbes are vertically transmitted is unknown.

(Line 123) I would remove "at least partially". If the patterns are just "partially" explained by vertical transmission, you can (and you did) mention and discuss later.

(Line 158) Please, include here how much these "96 Swarm OTUs and 103 97% OTUs" represent from the total number of OTUs for each approach. This will help to understand the phrase in line 167 "these small numbers of "shared" OTUs" and in line 270 "The microbiota of the Ariamnes spiders showed a low proportion of core OTUs".

(Line 159) resp. = respective? Would be better to include the complete word. Also, this percentage refers to shared or total OTUs?

(Line 160) Specify what are the sequences being compared for the "segregating site".

(Line 161) Describe what is a shared OTU in this study before mentioning "These "shared" OTUs".

(Line 165) Is that information right "except 2 Swarm OTUs out of 132", or should it be 96 instead 132?

(Figure 2, panel b) There are (i) two times - change if incorrect. Also, if you won't say anything about II (that is missing) or IV, you can remove this detail in the figure.

(Line 166) Include a short definition of the "global type-I error of HOME".

(Line 167) Simple dot?

(Line 175) Remove "any".

(Line 262) Please, specify the supplementary results.

(Line 278) This rationale is not very clear, you may elaborate more on it - since other OTUs seem to have similar patterns in the figure you cited (Fig. 2, specifically panel A).

(Line 293) Remove "(many)", since you argue here that no symbiont is vertically transmitted.

(Line 305) Can the sequencing of different 16S variable regions (not V1-V3 as here) lead to different results using HOME? If that's the case, you can briefly mention here.

This manuscript uses sequencing data from *Ariamnes* spiders and its microbiota - displaying phyllosymbiosis pattern - to present an tool (HOME) developed by them that solve the problem of distinguish symbionts vertically transmitted from the environmentally acquired - since both can lead to this pattern. In addition to presenting an example of phyllosymbiosis pattern caused by non-vertically transmitted symbionts, the authors perform simulations that confirm the resultus. They show the HOME model is sensible enough to detect vertical transmission even with a low number of segregating sites - as may be expected in 16S rRNA data normally used in microbiome studies. I think this manuscript is important for two main reasons. First, for confirming that patterns of phyllosymbiosis can be caused by symbionts maintained in populations in different ways (vertically transmitted or environmentally acquired). Second, because once a phyllosymbiosis pattern is observed - what seems to be quite prevalent in several evolutionary lineages - one obvious next question is how these symbionts are being maintained/acquired. Selection will act on microbially influenced host phenotypes that are heritable, and it can be directly on the symbiont or on a host trait that influences the environmental acquisition.

The authors were able to take advantage of an interesting dataset for their goal, the paper is well presented, concise and to the point. This manuscript will be of interest to those working directly with phyllosymbiosis hypotheses, but also to the great mass of researchers currently working with microbiome data - as for those working with non-model organisms for which is more complicated to experimentally test modes of symbiont transmission.

I do not have any major concerns with the paper but have several minor recommendations for improving the clarity of the paper.

(Line 26) Include the word “vertically” when talking about vertical transmission.

(Line 48) It is the host that filters, and that is why we observe phyllosymbiotic pattern. Thus, I would suggest changing “including environmental filtering“ to “including host-filtering environmental acquired microbes”.

(Line 49) “But which processes drive phyllosymbiosis in nature remains unknown” you mentioned can be both, right? So it seems the problem is distinguishing it. I would suggest: “But distinguishing the processes driving phyllosymbiosis in nature remains challenging”.

(Line 88) For me it is not very clear why tree congruence is not expected when symbionts are environmentally acquired on a dependence on host-filtering. Also, the cited reference (12) actually shows that even when symbionts are acquired from the environment it can lead to mirrored phylogenies, due to host filtering. You mentioned this in discussion as well (Line 314).

(Line 129) These *Ariamnes* spiders are non-model organisms and whether or not their associated microbes are vertically transmitted is unknown.

(Line 123) I would remove “at least partially”. If the patterns are just “partially” explained by vertical transmission, you can (and you did) mention and discuss later.

(Line 158) Please, include here how much these “96 Swarm OTUs and 103 97% OTUs“ represent from the total number of OTUs for each approach. This will help to understand the phrase in line 167 “these small numbers of “shared” OTUs” and in line 270 “The microbiota of the *Ariamnes* spiders showed a low proportion of core OTUs”.

(Line 159) resp. = respective? Would be better to include the complete word. Also, this percentage refers to shared or total OTUs?

(Line 160) Specify what are the sequences being compared for the “segregating site”.

(Line 161) Describe what is a shared OTU in this study before mentioning “These “shared” OTUs”.

(Line 165) Is that information right “except 2 Swarm OTUs out of 132”, or should it be 96 instead 132?

(Figure 2, panel b) There are (i) two times - change if incorrect. Also, if you won't say anything about II (that is missing) or IV, you can remove this detail in the figure.

(Line 166) Include a short definition of the “global type-I error of HOME”.

(Line 167) Simple dot?

(Line 175) Remove “any”.

(Line 262) Please, specify the supplementary results.

(Line 278) This rationale is not very clear, you may elaborate more on it - since other OTUs seem to have similar patterns in the figure you cited (Fig. 2, specifically panel A).

(Line 293) Remove “(many)”, since you argue here that no symbiont is vertically transmitted.

(Line 305) Can the sequencing of different 16S variable regions (not V1-V3 as here) lead to different results using HOME? If that's the case, you can briefly mention here.

Re: mSystems01104-21 (Limited evidence for microbial transmission in the phylosymbiosis between Hawaiian spiders and their microbiota)

Dear Dr. Benoit Perez-Lamarque:

Thank you for submitting your manuscript to mSystems. We have completed our review and I am pleased to inform you that, in principle, we expect to accept it for publication in mSystems. However, acceptance will not be final until you have adequately addressed the reviewer comments.

Editor Notes (from Sarah):

This is a very nice manuscript and I look forward to seeing it in press. In addition to addressing the review comments, I would like to see the following addressed:

> Thank you for your positive comments. Please see below a point-by-point response to your comments and the review comments.

(1) Fig. 1 - should the internal branches of (A) be colored as they are or black as in (B)? Also, could you please mark which nodes on the trees support phylosymbiosis? I think it's helpful to readers to put a circle or star on the congruent nodes (so they do not have to search for them).

> We have updated Fig. 1 accordingly. The branches of the phylogenetic tree of the host spiders (A) and those of the microbiota dendrogram (B) are colored by areas if all the descending tips come from the same geographic area. In addition, we have put circles on the nodes of the host phylogeny that significantly support phylosymbiosis: to identify these nodes we have used clade-specific Mantel tests (following Perez-Lamarque *et al.*, 2021). We have explained this methodology in the Methods section and in Supplementary Methods 2. Following the suggestion of reviewer #1, we have also changed the colors of the areas to make them colorblind-friendlier.

(2) I agree with reviewer 1's point about the colors in fig. 2a being too similar; I recommend changing the colors and possibly adding a shape to make it easier to see (instead of all points being circles).

> We have now removed the colors and instead only use 4 different shapes to differentiate the 4 kinds of OTUs. We have only retained the (i) "orange" and (ii) "purple" colors to indicate (i)

the OTUs that reject the null hypothesis of independent evolution in HOME and (ii) the endosymbionts respectively.

(3) I also had a question similar to reviewer 1 about the justification of using OTUs over ASVs.

> Our method HOME uses the variation within an OTU to infer whether or not this OTU has been vertically transmitted. In other words, the representative sequences of this OTU across all the host lineages have to show a certain variability (at least one mutation) to be able to infer its evolutionary history. By using ASV, we won't have access to such variation, because ASVs are supposed to be unique sequences, and variations within/beyond ASV are expected to be only sequencing/PCR errors. Therefore, without any "within-ASV variation", our method HOME cannot be run. One possibility would be to cluster "closely-related ASVs" and investigate whether these "closely-related ASVs" have been vertically transmitted. This would be quite similar to what we are doing: first, we did a 97% OTU clustering or a Swarm clustering (resulting in within-OTU variation), and second, for each OTU and each host lineage, we took the most (or the second most) abundant sequence as a representative, such that sequencing/PCR errors (that are expected to be in minority) should be discarded.

We have clarified this in the text:

- in the Results section (lines 165-167): "Given that HOME uses the intra-OTU variation to detect vertically transmitted OTUs, HOME can only be run on OTU alignments having at least one segregating site."

- in the Methods section (lines 437-439): "because HOME uses the intra-OTU variation, HOME cannot be run on ASVs (amplicon sequence variant) that are unique sequences (obtained after removing sequencing/PCR errors) and therefore present no "intra-unit" variation. HOME was therefore only run on Swarm and 97% OTUs".

We could use ASVs to test for phyllosymbiosis, however this would make our analyses inconsistent. We actually used ASVs in Armstrong *et al.* (2020), and we also found a significant phyllosymbiosis. We have added this information lines 413-414: "Similar trends using ASVs (amplicon sequence variants) instead of OTUs were reported in (17), so we did not replicate the analyses using ASVs here."

Reviewer comments:

Reviewer #1 (Comments for the Author):

Thank you for the opportunity to review this paper, I thoroughly enjoyed reading it. The authors address an interesting question regarding the patterns that regulate phylosymbiosis and take a novel approach to disentangle whether vertical transmission or other processes may be responsible for these patterns. The authors laid out their predictions clearly, and I felt that they did a good job explaining the function of their model in the context of their data. The development of a new computational tool to address questions regarding microbial-host coevolution and phylosymbiosis is likely to be of interest to many readers.

> Thank you very much for these positive comments.

In general, I have a few comments I would like the authors to address:
1.) What is the justification for using OTUs instead of ASVs in your analyses? I appreciate the comparison of using swarm and 97% OTU clustering, but I'm interested in why you avoided using ASVs.

> See above for a complete response. In short, given that HOME uses the intra-OTU variation to detect vertically transmitted OTUs, HOME can only be run on OTU alignments having at least one segregating site, therefore it cannot be run with ASVs, which do not contain “intra-ASV variation” other than sequencing/PCR errors.

2.) While I understand that the focal interest of this research team is the Ariamnes species complex in Hawaii, I'm wondering if it wouldn't bolster the paper to conduct a similar test of HOME using information from additional lineages of spiders or arthropods. There are a number of difficulties your system presents for your model, as you mentioned throughout the manuscript, and I think it would be useful to see another case study. If not that, a further explanation of why things like endosymbionts might not appear to be vertically transmitted here. What about these spiders might explain this pattern?

> We would love to conduct similar tests in other arthropods clades, however we are not aware of other available datasets: we would need metabarcoding datasets characterizing the microbiota of a monophyletic clade of arthropods sampled and processed in similar conditions. We hope that our analyses will motivate future efforts to acquire such data. We have now discussed that lines 319-320.

Endosymbionts can be conserved over long timescales (Degnan *et al.*, 2004; Bailly-Bechet *et al.*, 2017) but not necessarily so: there are examples of endosymbionts horizontally transmitted at a high rate (Baldo *et al.*, 2008). The processes that influence the rate of endosymbiont horizontal transmission remain unclear. For *Ariamnes* spiders, we can speculate that predation on other arthropods and cannibalism facilitate endosymbionts horizontal transfer. We have added this potential explanation in lines 343-348.

3.) Also, given the geographic and phylogenetic structure of your study system, how can you confirm the validity of the model's ability to differentiate between phylogenetically- and geographically-driven host switches?

> Actually, we don't say that our model can differentiate between phylogenetically- and geographically-driven host switches in this system. Instead, because of the strongly correlated geographic and phylogenetic structure, both phylogenetically- or geographically-driven host-switches leave similar signals in the microbial alignments and are therefore undifferentiable. We have now clarified this in the Discussion (lines 257-260) and in the Supplementary Results 1.

4.) While I liked the visualizations of the data, they could be improved by using a colorblind friendlier palette. In particular, the colors used for Molokai and Kohala in Figure one are very similar, and the colors used in Figure 2a are quite difficult to differentiate.

> We have now changed the colors of Figure 1 and used more contrasting colors to make it colorblind friendlier. In addition, we have removed the 4 colors in Figure 2a and replaced them with 4 different shapes.

Reviewer #2 (Comments for the Author):

This manuscript uses sequencing data from Ariamnes spiders and its microbiota - displaying phylosymbiosis pattern - to present an tool (HOME) developed by them that solve the problem of distinguish symbionts vertically transmitted from the environmentally acquired - since both can lead to this pattern. In addition to presenting an example of phylosymbiosis pattern caused by non-vertically transmitted symbionts, the authors perform simulations that confirm the resultus. They show the HOME model is sensible enough to detect vertical transmission even with a low number of segregating sites - as may be expected in 16S rRNA data normally used in microbiome studies. I think this manuscript is important for two main reasons. First, for confirming that patterns of phylosymbiosis can be caused by symbionts maintained in populations in different ways (vertically transmitted or environmentally acquired). Second, because once a phylosymbiosis pattern is observed - what seems to be quite prevalent in several evolutionary lineages - one obvious next question is how these symbionts are being maintained/acquired. Selection will act on microbially influenced host phenotypes that are heritable, and it can be directly on the symbiont or on a host trait that influences the environmental acquisition.

The authors were able to take advantage of an interesting dataset for their goal, the paper is well presented, concise and to the point. This manuscript will be of interest to those working directly with phylosymbiosis hypotheses, but also to the great mass of researchers currently working with microbiome data - as for those working with non-model organisms for which is more complicated to experimentally test modes of symbiont transmission.

> Thank you very much for these positive comments.

I do not have any major concerns with the paper but have several minor recommendations for improving the clarity of the paper.

(Line 26) Include the word "vertically" when talking about vertical transmission.

> Done.

(Line 48) It is the host that filters, and that is why we observe phylosymbiotic pattern. Thus, I would suggest changing "including environmental filtering" to "including host-filtering environmental acquired microbes".

> Done. We have replaced it by “including host-filtering of environmentally acquired microbes”.

(Line 49) "But which processes drive phylosymbiosis in nature remains unknown" you mentioned can be both, right? So it seems the problem is distinguishing it. I would suggest: "But distinguishing the processes driving phylosymbiosis in nature remains challenging".

> Agreed. Thanks for this suggestion!

(Line 88) For me it is not very clear why tree congruence is not expected when symbionts are environmentally acquired on a dependence on host-filtering. Also, the cited reference (12) actually shows that even when symbionts are acquired from the environment it can lead to mirrored phylogenies, due to host filtering. You mentioned this in discussion as well (Line 314).

> We rephrased this paragraph in the introduction to clarify the differences between “phylosymbiotic” and “cophylogenetic” patterns. Ref. (12) explains that if host-filtering might generate a phylosymbiotic pattern, it generally doesn’t create a cophylogenetic pattern, that is mainly generated through vertical transmission. Nevertheless, we agree that in some (rare) conditions, host-filtering can lead to congruent phylogenies, but this is an exception. We have therefore rephrased our sentence: “Conversely, in the absence of vertical transmission, such [phylogenetic] congruence may only occur in very specific cases, but it is not expected in general (12)”.

(Line 129) These Ariamnes spiders are non-model organisms and whether or not their associated microbes are vertically transmitted is unknown.

> Done.

(Line 123) I would remove "at least partially". If the patterns are just "partially" explained by vertical transmission, you can (and you did) mention and discuss later.

> Done.

(Line 158) Please, include here how much these "96 Swarm OTUs and 103 97% OTUs" represent from the total number of OTUs for each approach. This will help to understand the phrase in line 167 "these small numbers of "shared" OTUs" and in line 270 "The microbiota of the Ariamnes spiders showed a low proportion of core OTUs".

> Done. They represent 23% (for Swarms OTUs) and 25% (for OTUs at 97%) of the total OTUs.

(Line 159) resp. = respective? Would be better to include the complete word. Also, this percentage refers to shared or total OTUs?

> The “resp.” referred to the comparison between the Swarms OTUs and the OTUs at 97% (i.e. OTUs defined with a threshold of 97%); the percentage here just refers to the clustering algorithm. We have clarified this sentence and removed “resp”: “only 51 Swarm OTUs and 66 OTUs at OTUs at 97% had at least one segregating site in the OTU alignment, while we had 81 Swarm OTUs and 90 OTUs at 97% when selecting the second most abundant sequence.”

(Line 160) Specify what are the sequences being compared for the "segregating site".

> We now indicate “... at least one segregating site in the OTU alignment”.

(Line 161) Describe what is a shared OTU in this study before mentioning "These "shared" OTUs".

> Done, we have added the sentence: “...OTUs that were shared by at least 5 spider individuals (we thereafter refer to these OTUs as the “shared OTUs”, in opposition to “unshared OTUs” that are detected in less than 5 spider individuals).”

(Line 165) Is that information right "except 2 Swarm OTUs out of 132", or should it be 96 instead 132?

> The 132 Swarms OTUs correspond to the addition of the 51 Swarm OTUs obtained when selecting the most abundant sequence per host individual and the 81 Swarm OTUs obtained when selecting the second most abundant sequence per host individual. We have clarified this.

(Figure 2, panel b) There are (i) two times - change if incorrect. Also, if you won't say anything about II (that is missing) or IV, you can remove this detail in the figure.

> Sorry for the typo, we have replaced the second “i” by “ii”.

(Line 166) Include a short definition of the "global type-I error of HOME".

> We have now defined it as the percentage of environmentally acquired OTUs that are incorrectly inferred as being vertically transmitted by the model.

(Line 167) Simple dot?

> Done.

(Line 175) Remove "any".

> Done.

(Line 262) Please, specify the supplementary results.

> We now refer to them as **Supplementary Results 1**.

(Line 278) This rationale is not very clear, you may elaborate more on it - since other OTUs seem to have similar patterns in the figure you cited (Fig. 2, specifically panel A).

> We have clarified this: “The two OTUs that rejected the null hypothesis of independent evolution with HOME, which respectively belong to the ubiquitous *Bacillus* and *Erythrobacter* genera, have high estimated numbers of host-switches, which likely resulted in incongruent cophylogenetic patterns (Fig. 2). In addition, their estimated parameters fit into the distribution of parameters obtained for the non-transmitted OTUs (Fig. 2). Thus, we can conclude that these OTUs are likely false positives.”

(Line 293) Remove "(many)", science you argue here that no symbiont is vertically transmitted.

> Done.

(Line 305) Can the sequencing of different 16S variable regions (not V1-V3 as here) lead to different results using HOME? If that's the case, you can briefly mention here.

> Yes, the faster the DNA marker region evolves, the more information we have for detecting vertically transmitted symbionts. We now write: “Future works specifically targeting microbial marker genes or another 16S rRNA region that are longer or evolve faster...”.

References:

Armstrong EE, Perez-Lamarque B, Bi K, Chen C, Becking LE, Lim JY, Linderoth T, Krehenwinkel H, Gillespie R. 2020. A holobiont view of island biogeography: Unraveling patterns driving the nascent diversification of a Hawaiian spider and its microbial associates. *bioRxiv*.

Bailly-Bechet M, Martins-Simões P, Szöllösi GJ, Mialdea G, Sagot M-FF, Charlat S. 2017. How long does Wolbachia remain on board? *Molecular Biology and Evolution* **34**: 1183–1193.

Baldo L, Ayoub NA, Hayashi CY, Russell JA, Stahlhut JK, Werren JH. 2008. Insight into the routes of Wolbachia invasion: High levels of horizontal transfer in the spider genus *Agelenopsis* revealed by Wolbachia strain and mitochondrial DNA diversity. *Molecular Ecology* **17**: 557–569.

Degnan PH, Lazarus AB, Brock CD, Wernegreen JJ. 2004. Host-symbiont stability and fast evolutionary rates in an ant-bacterium association: Cospeciation of *Camponotus* species and their endosymbionts, *Candidatus blochmannia*. *Systematic Biology* **53**: 95–110.

Perez-Lamarque B, Maliet O, Selosse M-A, Martos F, Morlon H. 2021. Do closely related species interact with similar partners? Testing for phylogenetic signal in bipartite interaction networks. *bioRxiv*: 2021.08.30.458192.

December 16, 2021

Dr. Benoit Perez-Lamarque
IBENS
Paris
France

Re: mSystems01104-21R1 (Limited evidence for microbial transmission in the phylosymbiosis between Hawaiian spiders and their microbiota)

Dear Dr. Benoit Perez-Lamarque:

Your manuscript has been accepted, and I am forwarding it to the ASM Journals Department for publication. For your reference, ASM Journals' address is given below. Before it can be scheduled for publication, your manuscript will be checked by the mSystems senior production editor, Ellie Ghatineh, to make sure that all elements meet the technical requirements for publication. She will contact you if anything needs to be revised before copyediting and production can begin. Otherwise, you will be notified when your proofs are ready to be viewed.

Publication Fees:

We recognize that the video files can become quite large, and so to avoid quality loss ASM suggests sending the video file via <https://www.wetransfer.com/>. When you have a final version of the video and the still ready to share, please send it to mssystemsjournal@msubmit.net.

Sincerely,

Sarah Hird
Editor, mSystems

Journals Department
Phone: 1-202-942-9338